# Performance of an Adaptive Aggregation Mechanism in a Noisy WLAN Downlink MU-MIMO Channel

**Lemlem Kassa** [1,*], **Mark Davis** [2], **Jianhua Deng** [1] **and Jingye Cai** [1]

1   School of Information and Software Engineering, University of Electronic Science and Technology China (UESTC), Chengdu 610054, China; jianhua.deng@uestc.edu.cn (J.D.); jycai@uestc.edu.cn (J.C.)
2   Communication Network Research Institute (CNRI), Technological University Dublin, D08 NF82 Dublin, Ireland; mark.davis@tudublin.ie
*   Correspondence: lemlem.kassa@aastu.edu.et

**Abstract:** This paper investigates an adaptive frame aggregation technique in the medium access control (MAC) layer for the Wireless Local Area Network (WALN) downlink Multi-User–Multiple-In Multiple-Out (MU-MIMO) channel. In tackling the challenges of heterogeneous traffic demand among spatial streams, we proposed a new adaptive aggregation algorithm which has a superior performance over the baseline First-in–First-Out (FIFO) scheme in terms of system throughput performance and channel utilization. However, this earlier work does not consider the effects of wireless channel error. In addressing the limitations of this work, this study contributes an enhanced version of the earlier model considering the effect of channel error. In this approach, a dynamic adaptive aggregation selection scheme is proposed by employing novel criteria for selecting the optimal aggregation policy in WLAN downlink MU-MIMO channel. Two simulation setups are conducted to achieve this approach. The simulation setup in Step 1 performs the dynamic optimal aggregation policy selection strategy as per the channel condition, traffic pattern, and number of stations in the network. Step 2 then performed the optimal wireless frame construction that would be transmitted in the wireless channel in adopting the optimal aggregation policy obtained from Step 1 that maximizes the system performance. The proposed adaptive algorithm not only achieve the optimal system throughput in minimizing wasted space channel time but also provide a good performance under the effects of different channel conditions, different traffic models such as Pareto, Weibull, and fBM, and number of users using the traffic mix of VoIP and video data. Through system-level simulation, our results again show the superior performance of our proposed aggregation mechanism in terms of system throughput performance and space channel time compared to the baseline FIFO aggregation approach.

**Keywords:** channel error; frame size adaptation; heterogeneous traffic; MU-MIMO wireless local area networks (WLANs); wasted space channel time

## 1. Introduction

Due to the rapid growth of wireless users and traffic demand in communication network, Wireless Local Area Network (WLAN) development has introduced different technologies to support the performance of the network. MAC layer aggregation is among the technologies introduced in the IEEE 802.11 n/ac standards to accommodate the increasing traffic demand in current WLANs [1–3] to support aggregation of multiple frames directing to the same station (STA) into a single wireless frame. This allows for a reduction in the signaling overhead per frame transmission. Moreover, to accommodate the huge demand of users' requests and data transmission speeds in WLAN, the IEEE 802.11 ac standard also provides Multiple-User Multiple-Input Multiple-Out (MU-MIMO) technology at the physical layer. This technology enables the parallel transmission of multiple user frames at a time, forming virtual spatial transmission channels between the AP and

the multiple receiving STAs [4–6]. Moreover, the combination of MU-MIMO and MAC layer frame aggregation approaches can significantly improve the system performance of WLAN [5,6]. However, due to the heterogeneous traffic demand among spatial streams, the advantage of multiuser (MU) data transmission is not always achievable in downlink MU-MIMO transmission. In the WLAN downlink MU-MIMO channel, a group of multiplexed frames are constructed such that all users' frames have an identical transmission duration. However, if the data transmission duration among spatial streams is different, 'space channel time' will occur [5,7–11]. Space channel time is a time duration where some of the streams carry user data while the others do not.

Although MU frame construction is significant, the problem has not been explored much in the literature. For example, ref. [11] provides a frame aggregation scheme to reduce space channel time and system throughput performance in the WLAN downlink MU-MIMO channel. However, these approaches only consider VoIP traffic, and the effect of channel error is not studied. Space channel time can be avoided by adding frame padding dummy bits at the tail of the shorter frame until it has the same transmission duration as the others in downlink MU-MIMO transmission. However, this approach leads to degradation in transmission efficiency [8,9,12,13]. Emphasizing the challenges of heterogeneous traffic demand and transmission bit rate among stations, ref. [8] proposed an analytical scheme to construct a MU frame with an optimal length to maximize transmission efficiency for both downlink and uplink transmissions. However, the optimal wireless frame setting in downlink MU-MIMO transmission is not only affected by the challenges of heterogeneous traffic demand and transmission bit rate, but also channel condition, which is not addressed in this study.

The system throughput performance of MU-MIMO transmission can be increased by reducing signaling overhead and frame error rate to enhance the transmission performance. Nonstationary devices often experience path loss, thermal noise, interference, and fading in signal-to-noise ratio due to their nature of mobility [14–16], which affects the throughput performance in WLAN. Frame size is one of the main performance factors that directly impact the communication efficiency of WLAN in terms of useful transmitted data and overhead introduced by headers and preambles [3,9,17–20]. If the wireless frame size is very large, a bit error can destroy the whole frame, as there is no forward error correction mechanism that allows an increase in the frame success rate and an improvement in the throughput performance [17–20]. On the other hand, if the frame size is small the overhead frames, such as MAC and PHY headers, interframe spacing, acknowledgment, and backoff timer would occupy a large portion of the transmission time, thus reducing the transmission efficiency [9,17–20].

Some frame aggregation algorithms have been proposed by [17–20] to improve the performance in error-prone wireless networks by determining the optimal frame size under the current condition of the channel. However, these approaches cannot be accessible in the WLAN MU-MIMO channel. An efficient frame aggregation scheme is proposed by [9], improving system performance and reducing space channel time in the WLAN downlink MU-MIMO channel. A hybrid Aggregate MAC Service Data Unit (A-MSDU/) Aggregate MAC Protocol Data Unit (A-MPDU) aggregation scheme was employed, which achieves the frame error rate by using the A-MSDU frame size adaption technique. However, the wireless frame aggregation setting is not adaptive to the change in heterogeneous traffic patterns.

The main contribution of this paper is to achieve an efficient adaptive frame aggregation algorithm for downlink MU-MIMO transmission in terms of system throughput performance and space channel time by extending our previous work [11]. Since the performance of different aggregation policies perform differently under the dynamic effects of traffic patterns and channel conditions, the determination of the optimal aggregation policy which provides the maximum system performance is challenging. To tackle these challenges, we achieved the simulation setup in two steps. Step 1 performs the dynamic optimal aggregation policy selection strategy as per the channel condition, traffic pattern,

and number of stations in the network. Step 2 then performs the optimal wireless frame construction to adopt the optimal aggregation policy obtained from Step 1. Finally, MU-MIMO transmission is achieved. A novel criterion is employed in the proposed approach to select the optimal frame aggregation policy, which allows the optimal wireless frame setting to be determined. The detailed description of the proposed approach is elaborated in Section 4. In this study, a system-level simulation is conducted using MATLAB programming language and the performance of the proposed approach is evaluated considering the effects of various channel conditions, traffic models (such as Pareto, Weibull, and fractional Brownian Motion (fBM) [11]), and the number of stations as compared to the baseline FIFO aggregation algorithm.

The remainder of this paper is outlined as follows: Section 2 discusses a review of frame aggregation schemes on the performance of the WLAN downlink MU-MIMO channel and our motivation. Section 3 discusses the architecture of our WLAN system and the technical components used at the MAC layer and PHY layer according to the IEEE 802.11 ac specifications. Section 4 discusses the proposed approach. The details of the experimental results and discussion will then be explained in Section 5. Finally, conclusions are given in Section 6.

## 2. Related Work and Our Motivation

Frame aggregation is one of the techniques that affect the performance of WLAN. In this section, some previous studies on frame aggregation techniques for the system performance of the WLAN downlink MU-MIMO channel are reviewed.

### 2.1. Related Work

Moriyama et al. [7] investigated a frame aggregation size determination strategy for the WLAN downlink MU-MIMO channel by considering channel utilization and delay. This approach proposed different aggregation models to determine aggregation frame size to consider the effect of the traffic variation among spatial streams. The basic principle of this scheme is to determine the aggregation frame size dynamically between minimum queue length and average queue length, according to the variation of queue lengths among streams. To implement this scheme, the throughput variation of each stream was evaluated by employing a timestamp associated with data frames when the data frames arrived into the queue. The presented results indicate high channel efficiency in minimizing transmission delay and improved channel utilization. However, the proposed approach is not evaluated under the effect of channel error, and the traffic model used is not discussed.

In analyzing the impacts of different overhead components in WLAN multi-user transmission, a data frame construction scheme called DFSC is proposed by [8] to maximize the transmission efficiency by evaluating the optimal length according to the status of buffers and transmission rates in both uplink and downlink multiuser transmissions. However, this scheme does not consider the effect of channel error, which could reduce the transmission performance due to excessive retransmissions of frames received in error. Moreover, the assumption of the Poisson traffic model used for packet arrival distribution is insufficient to characterize the actual traffic scenarios.

A frame-duration-based frame aggregation scheme was proposed by Nomura et al. [9]. A receiving mobile terminal (MT) selection strategy was proposed, which provides high priority to STAs with higher throughput in the next MU-MIMO and has a large amount of data in their buffer. In this approach, the wireless frame setting is limited in all spatial streams considering their Modulation Coding Scheme (MCS) level. However, this approach always prioritizes stations with higher transmission rates. According to the presented results, an efficient frame aggregation scheme is proposed which minimizes space channel time, enhances system throughput, and decreases frame error rate. Moreover, the limitation of this study is the assumption of a Poisson traffic model to characterize packet arrivals, which is an unrealistic model, and the performance of the proposed approach is not evaluated under the effects of heterogenous traffic scenarios.

A frame-size-based aggregation scheme was proposed by [10] to reduce the space channel time in the downlink MU-MIMO channel. The basic principle of this approach was to use a uniform data frame size in each spatial stream by aggregating an equal number of frames under a constant data rate. The study considered a Poison traffic model, which is not adequate to represent the real network traffic scenario. In addition to this, a uniform data frame size aggregation policy cannot always achieve better performance due to the heterogeneous traffic nature among spatial streams, and the effect of channel error was ignored in the study, thus the algorithm is not applicable.

A dynamic adaptive aggregation selection algorithm was proposed by [11]. This approach employed different aggregation policies to determine the optimal aggregation frame size on each spatial stream. According to the simulation results, superior performance over the baseline FIFO algorithm in terms of system throughput performance and channel utilization was achieved by tackling the challenges of heterogeneous traffic demand among special streams. Different traffic models for VoIP communication were adopted to evaluate the performance of the proposed approach. However, the study was carried out under the assumption of ideal channel condition, which is unrealistic according to the real condition of the WLAN channel, thus resulting in a suboptimal solution. For more realistic results, this study will extend the limitation of this work to improve the system performance by considering the effect of channel error, which has a significant role in the determination of the optimal frame aggregation size before transmission. Focusing on the padding problem of downlink MU-MIMO, some research work has been carried out by [12,13] mainly to increase the transmission efficiency of multiuser (MU) frames and minimize space channel time. The proposed scheme replaces padding bits with the data frame of other stations. However, the effects of channel error and traffic model utilized are not considered.

### 2.2. Motivation of This Work

The dynamic adaptive frame aggregation selection scheme can maximize the system throughput performance of WLAN while enhancing channel utilization and transmission efficiency. However, this approach was conducted under the assumption of ideal channel conditions, which is unrealistic according to the real condition of the error-prone wireless channel. The motivation of this work comes from the above-mentioned drawback of [11] and we aim to provide an efficient dynamic adaptive aggregation selection algorithm that can mitigate drawbacks for downlink MU-MIMO transmission in error-prone WLAN.

### 3. WLAN Architecture and Its Assumption

The system model is based on the IEEE 802.11 ac WLAN network, which is composed of a single central access point (AP) and four stations. In this topic, the assumptions of network topology and channel access models are discussed.

### 3.1. Network Topology Model

The network topology consists of a single AP with four antennas located in a fixed position at the center of the network. Since user selection is not considered in this study, a maximum of four stations are assumed which are randomly located from the AP. Both the AP and the STAs are stationary. Although some STAs may be equipped with multiple antennas, in this study we focused on stations with a single antenna. The AP can transmit independent data streams in parallel by using the MU-MIMO technique to the STAs up to the number of antenna ($N_{Ant} = 4$) at the AP, under the assumption of ideal channel orthogonality and ideal spatial channel separation between the spatial streams [9]. Figure 1 illustrates the network topology model.

We assume a fixed number of stations so that all operate in saturation condition, i.e., each STA always has a data frame available for transmission [17], since the particular focus of this study is frame-size-based to investigate the effect of heterogeneous traffic patterns among spatial streams to find out the optimal system frame size. Moreover, a constant data

rate of 260 Mbps is considered for each source STA, as the proposed aggregation scheme is frame-size-based.

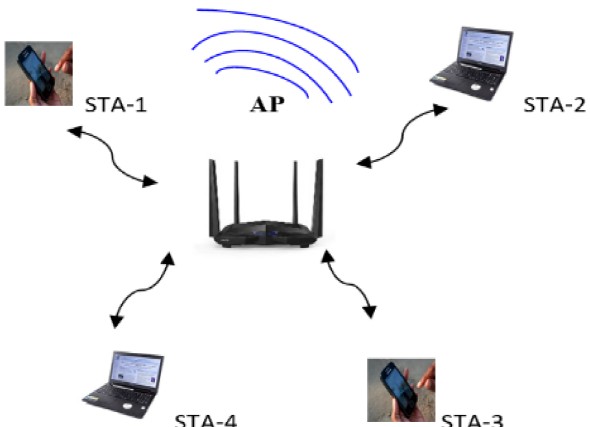

**Figure 1.** Network topology model.

### 3.2. Channel Access Model

The downlink MU-MIMO channel access procedure in this paper is assumed to be similar to that illustrated in Figure 2. The data transmission technique is performed according to the carrier-sense multiple access with collision avoidance (CSMA/CA) protocol, excluding the request to send (RTS) clear to send (CTS) procedures. Since data is only transmitted from the AP to the receiving STAs according to $Num_{STA} \leq N_{ANT}$, we consider the average backoff time, i.e., (CW_min-1)/2 × Tslot [18]. Therefore, in ideal channel conditions, after waiting for the idle time duration for Distributed Inter-Frame Space (DIFS) and back off time (BO), the AP simultaneously transmits data frames to each receiving STA through the MU-MIMO channel. If the data transmission time among spatial streams is different—for instance, STA—2, STA—3, and STA—4 as shown in Figure 2, wasted space channel time such as Space-1, Space-2, and Space-3 occurred. Space channel time is a time duration where part of spatial streams carry data, while the others do not. Frame padding bits generally added at the tail of the shorter stream until it has the same transmission duration as the others, however, this approach degrades the transmission efficiency. According to our assumptions, data frames are aggregated using any one of the aggregation policies, such as All Agg FA (Baseline Approach), Equal Frame Size FA, Equal MPDUs Agg FA, or Avg Num MPDUs FA adopted from the algorithm in [11]. However, due to the dynamic effects of traffic patterns among spatial streams and channel conditions, a specific aggregation policy cannot always achieve maximum performance. To handle these challenges, a novel criterion of selecting the optimal aggregation policy is employed by adopting a dynamic adaptive aggregation selection scheme considering the effects of channel conditions and traffic patterns. Therefore, the optimal frame setting on each spatial stream that maximizes the system throughput can be achieved.

The time required for completing a single MU-MIMO transmission $T_{DL-MU-MIMO}$ is given by Equation (1):

$$T_{DL-MU-MIMO} = T\text{DIFS} + BOTime + \max(TData_i) + NumSTA(TSIFS + TBA) \quad (1)$$

where:

- *BOTime* is Backoff time;
- $PSDU\text{-}STA_i$ is the aggregated PSDU data frame of $STA_i$;
- $Space_{-i}$ is wasted space channel time of $STA_i$;
- $TData_i$ is data transmission time of the longer stream.

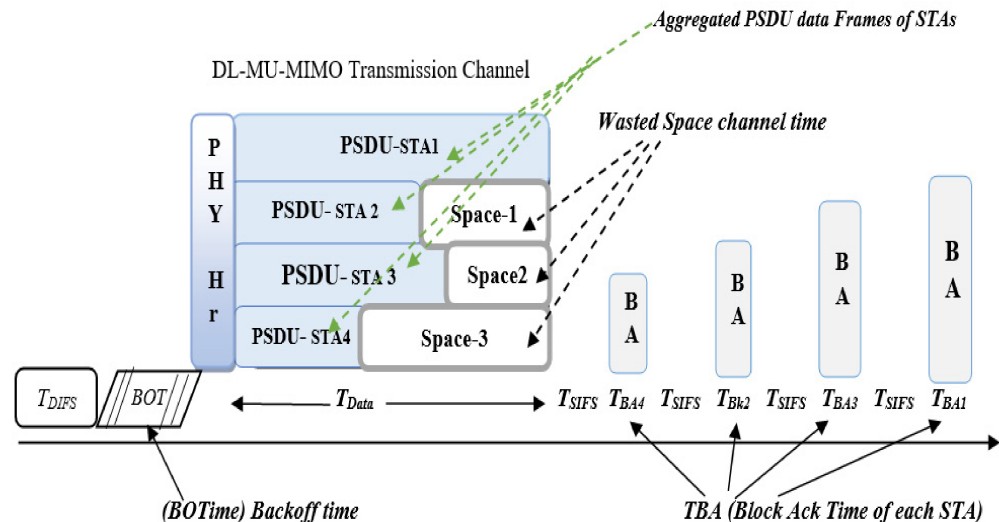

**Figure 2.** A diagram illustrates a single MU-MIMO transmission channel access for the case of 4 STAs.

The AP performs all MU transmission considering the error-prone wireless channel. The Additive White Gaussian Notice (AWGN) channel model is adopted in this study to examine the effects of different channel errors with varying signal-to-noise ratio (SNR) values. Because, according to our assumptions of ideal channel orthogonality and ideal spatial channel separation, AWGN channel modal is appropriate which can allow a line-of-sight communication between the transmitter and receiver [21,22]. In this channel model, SNR is used as a performance parameter for assessing the quality of communication [16,17], and it is specified as the ratio of signal power to noise power. In this regard, the higher the SNR value, the better the channel quality, so the frame error occurrence rate is lower. In contrast, the lower the SNR value, the worse the channel quality, which leads to a higher frame error rate [15,17].

As shown in Figure 2, each receiving STA replies with Block Acknowledgement ($B_{Ack}$) frames individually. Frames which are retransmitted will be sent back to the AP buffer for the next transmission, as long as the maximum retransmission count of $N_{Ret} = 3$ is not exceeded. One MU-MIMO transmission is completed following these procedures. Moreover, regardless of the channel status, all control frames are transmitted with a basic rate (minimum rate).

## 4. Proposed Approach

This section proposes a frame aggregation scheme which employs a dynamic adaptive aggregation selection mechanism for WLAN downlink MU-MIMO transmission. The proposed approach aims to increase the system throughput per MU-MIMO transmission while reducing space channel time, by determining the optimal aggregation policy in tackling the challenges of heterogeneous traffic demand among spatial streams. Traffic generation is the first operation to generate MAC Service Data Unit (MSDU) MAC data frames in byte; according to our assumption, a constant frame size of 100 bytes for VoIP and 1000 bytes for video traffic is considered. Traffic models, such as Pareto, Weibull, and fBM, are adopted from [11] to perform the traffic generation task to generate both VoIP and video traffics. The AP-Buffer Manager at the AP buffers each user's frames as long as there is space available to accommodate the incoming new frames. The maximum AP buffer size considered in this experiment is *BufAP* = 50 *MB*. If the buffer is full, the reception of new packets will be denied, and the AP suspends invoking the traffic generator and continues with the output process until some space becomes free at the buffer. Frames at the AP-Buffer Manager also accommodate retransmitted frames; remaining frames which are not selected for aggregation because of the maximum A-MPDU frame size limitation specified in the IEEE 801.11 ac standard; and remaining frames which are not selected

because of the maximum A-MPDUs size limitation allowed by the different aggregation policies, such as All Agg FA (Baseline Approach), Equal Frame Size FA, Equal MPDUs Agg FA, and Avg Num MPDUs FA [11]. The acronym FB in each aggregation approaches means 'frame aggregation'. The next operation of MAC layer aggregation is performed by the Aggregation Manager. We employed the two-layer hybrid MAC-layer aggregation methods, such as A-MSDU/A-MPDU to enhance transmission performance and minimize frame error rate while using A-MSDU, which allows the entire frame to be retransmitted if the receiving station finds it has transmission errors in the case of using only A-MSDU [9,20].

All Agg FA (Baseline Approach) is used as a baseline aggregation algorithm to compare the performance of the proposed adaptive aggregation algorithm. All Agg FA (Baseline Approach) follows a FIFO aggregation, which allows larger frame size aggregation as much as the maximum aggregation size is allowed per transmission. Equal Frame Size FA emphasizes the equalizing of the number of frames aggregated in all spatial streams to allow the aggregated frame length equal in all streams. In this aggregation approach, a station with the shorter frame size in its buffer always selected to determine the aggregated A-MPDU frame size of all the other STAs in the stream. Equal MPDUs Agg FA aggregation policy considers equal number of MPDUs to aggregate frames on each stream, and Avg Num MPDUs FA aggregation considers average number of MPDUs to aggregate frames on each stream [11]. Since the performance of different aggregation policies performs differently under the dynamic effects of traffic patterns and channel conditions, the determination of the optimal aggregation policy which provides the maximum system performance is challenging. To tackle these challenges, we perform the simulation setup in two steps: The simulation setup in Step 1 performs the optimal aggregation policy selection strategy under the different channel conditions (SNR = 3,10, or 20 dB), traffic models (Pareto, Weibull, or fBM), and number of STAs. Figure 3 illustrates the procedure in Step 1. Once the traffic is generated and buffered in Step 1, the Aggregation Manager aggregates the data frame of each user, employing the four different aggregation policies as depicted in [11], such as (Agg—1, Agg—2, Agg—3, and Agg—4), which represent (All Agg FA (Baseline Approach), Equal Frame Size FA, Equal MPDUs Agg FA, and Avg Num MPDUs FA) aggregation policies [11], respectively. Then, the aggregated data frames of each aggregation policy are transmitted one by one to the receiving STAs (Receivers) using the MU-MIMO technique, thus the receiving stations receives different aggregated frame size obtained from different aggregation polices. Then, the Performance Analyzer evaluates the performance of each aggregation policy, determining which provides the maximum system throughput performance, and records the optimal aggregation policy. Therefore, from all possible frame sizes obtained from the different aggregation policies, the optimal system frame size is the one obtained from the optimal aggregation policy.

Equation (2) illustrates how to identify the optimal system throughput and the corresponding optimal aggregation policy obtained from the simulation setup in Step 1. *SysThroughputMax$_t$* gives the maximum system throughput performance and *Index_Agg_Policy$_t$* is the index of the corresponding optimal aggregation policy at time *t*. Aggregation policies are assumed to be arranged in order of (1 to 4), respectively, for Agg_1, Agg_2, Agg_3, and Agg_4. Therefore, if the *Index_Agg_Policy$_t$* = 1, the maximum system throughput is achieved by AGG_1, which can achieve the maximum system throughput. Otherwise, if the *Index_Agg_Policy$_t$* = 4, it is AGG_4.

$$[SysThroughputMax_t, \; Index\_Agg\_Policy_t] = \max\left(SysTh_{Agg\_1}, SysTh_{Agg\_2}, Sys\, Th_{Agg\_3}, Sys\, Th_{Agg\_4}\right) \qquad (2)$$

The system throughput value of each aggregation policy can be computed using the expression in Equation (3). Therefore, according to our expression in Equation (2), *SysTh$_{Agg\_i}$* represented the system throughput obtained by the ith aggregation policy. Thus, the results obtained from Equation (2), such as *SysThroughputMax$_t$*, and *Index_Agg_Policy$_t$* are used as feedback to achieve the experiment in Step 2. System throughput is the average data rate at which the AP can successfully deliver to all receiving stations [9,11]. It can be computed as the ratio of the sum of all the successful frame sizes of the system over total

channel transmission time. It involves data transmission time, DIFS and Backoff times, and block acknowledgment time, as shown in Equation (1):

$$System\ throughput = \frac{\sum_{i=1}^{4} Succe\_FrameSize\_Stream_i}{T_{DL-MU-MIMO}} \tag{3}$$

where:

- $SysThroughputMax_t$ is the maximum system throughput;
- $Index\_Agg\_Policy_t$ is the index of the optimal aggregation policy;
- $T_{DL\_MU\_MIMO}$ is the total time required to complete a single MU-MIMO transmission according to the specification in Equation (1);
- $\sum_{i=1}^{4} Succe\_FrameSize\_Stream_i$ is the sum of all successful frames in all spatial streams. A maximum of four streams can be achieved according to our assumption.

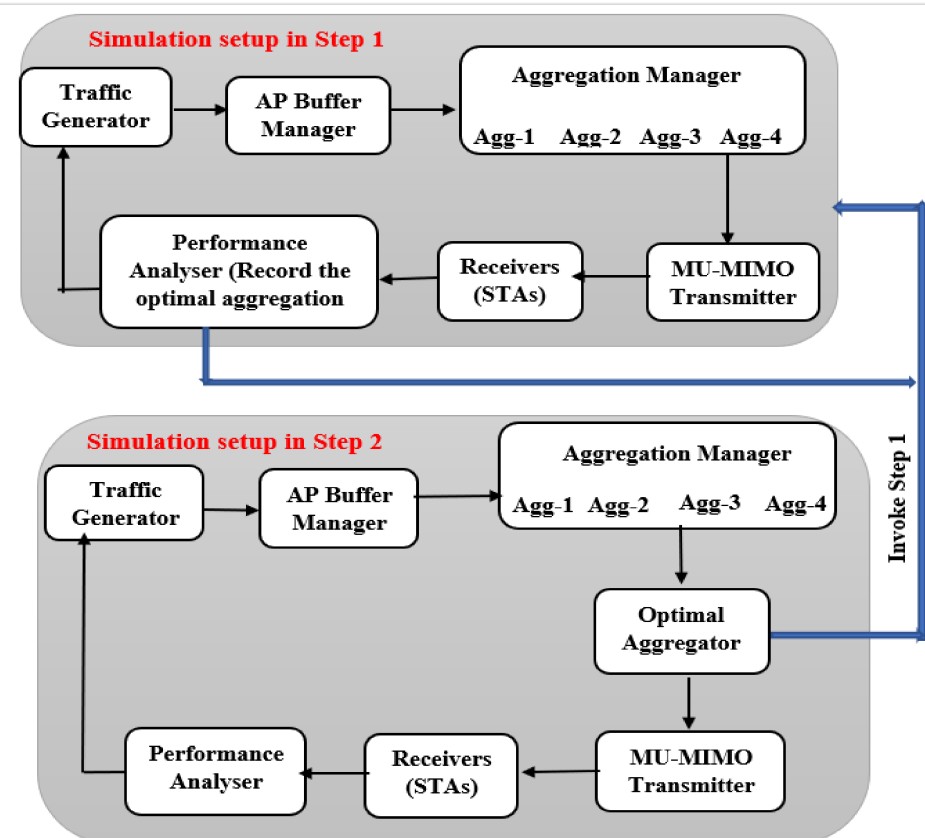

**Figure 3.** Interaction diagram for the simulation setups Step 1 and Step 2.

Then, as Figure 3 shows in simulation setup Step 1 after the AP identifies the optimal aggregation policy, it will be recorded and utilized in the future operation, i.e., every time a similar channel condition and traffic pattern and number of stations occurs in the system, the operations in Step 1 will not be executed again. Therefore, in simulation setup Step 2, frame aggregation can easily be constructed by the optimal aggregator. The process in the simulation setup of Step 2 is also demonstrated in Figure 3.

At this step, once the traffic is generated and buffered, the AP should first observe the channel condition and the traffic model in the network, then utilize the optimal aggregation policy provided by Step 1 for that particular channel condition and traffic model. Then, the optimal aggregator selects the optimal aggregation policy, which provides the optimal frame size of the system from the four provided aggregation policies by the Aggregation Manager. Following these procedures, the dynamic adaptive aggregation selection process is achieved throughout the simulation by aiming to realize the maximum system

performance. The Performance Analyzer in simulation setup Step 2 examines the optimal system throughput and minimum space channel time achieved in the system throughout the simulation time. The traffic generator, AP Buffer manager, and the Aggregation Manager perform similar tasks as specified in Step 1. However, in simulation setup Step 2 the optimal aggregation is known. The frame transmission is then performed by MU-MIMO Transmitter to the receiving STAs using the MU-MIMO technique.

## 5. Results and Discussion

In this section, the performance of the proposed approach is evaluated for downlink MU-MIMO transmission in WLAN through system-level simulation using MATLAB programming language. Simulation parameters are chosen for the IEEE 802.11 ac [1] standard, and the details of the system parameter settings are shown in Table 1.

**Table 1.** Simulation parameter.

| Parameters | Symbol | Value |
|---|---|---|
| Number of Antenna at AP | $N_{Ant}$ | 4 |
| Number of Stations | $Num_{STA}$ | 2–4 |
| Average Data Frame Length | LData | 100 Byte for VoIP, 1000 Byte for Video |
| Traffic Rate | | 10 Kbps for VoIP 100 Mbps for Video |
| Data Rate | | 260 Mbps per STAs |
| Minimum Window Size | CW-min | 15 |
| Transmission Time Slots | Tslot | 16 Microseconds |
| Average A-MSDU Length | | 11,454 Byte |
| Max Number of MPDU Frames Aggregated | | 64 |
| Max A-MPDU Length | | 1.0 Mbyte |
| SNR | | 3, 10, 20 dB |
| Max Number of Retransmission | $N_{Ret}$ | 3 |
| Buffer Size at the AP | BufAP | 50 MB |

### 5.1. Experimental Procedures

System throughput performance and space channel time ratio (%) under the effects of variable number of users, channel error, and different traffic models are examined in this section. System throughput is defined as the average data transmission rate at which the AP can successfully transmit to all receivers [9,11]. Space channel time is the ratio of total space channel time to the average time duration of one MU-MIMO transmission, i.e., the total space channel time plus data transmission time (see Equation (5)). Performance evaluation is carried out by comparing the proposed approach with All Agg FA (Baseline Approach). The AP can communicate with up to $Num_{STAs} \leq N_{ANT}$.

Rayleigh, Rician, and AWGN are among the basic channel models used to characterize channel behavior between transmitter and receiver. The AWGN channel model adopted in this study is used to examine the performance of the proposed approach under the effects of different channel conditions considering SNR = 3, 10, and 20 dB. The MATLAB function of $'awgn'$ is used in this analysis to model the AWGN channel. This channel model adopts white Gaussian noise on the transmitted signal to characterize the behavior of the received signal [21]. Equation (4) shows the expression of $'awgn'$ function in MATLAB programming.

$$Rsignal = awgn \left( Tsignal, SNR, 'measured' \right) \qquad (4)$$

where:

- *Tsignal* is transmitted signal with added white Gaussian noise;
- *Rsignal* is received signal with noise as per SNR in dB;
- *'measured'* specifies signal power [21];
- *SNR* is a signal-to-noise ratio.

The hybrid A-MSDU/A-MPDU frame aggregation strategy is adopted to reduce the transmission error, since in the case of A-MPDU aggregation the only retransmitted frame is the one with the error, unlike A-MSDU which retransmits the whole aggregated frame when an error has occurred in any one of the frames [2,9,18,20]. The error detection is performed on a per-MPDU basis in this study. The following experiments are conducted in this study: (i) performance of the system throughput under various channels' conditions; (ii) performance of the system throughput under a various number of users; (iii) performance of the system throughput under variable traffic data types; (iv) performance of the system throughput with increasing system traffic load; (v) performance of the space channel time ratio of the system.

### 5.2. Performance of System throughout under Different Channel Conditions and Number of Stations

Figures 4 and 5 demonstrate the system throughput performance under the effects of channel conditions and number of users, respectively. Figure 4 shows the performance of system throughput under various channel conditions, fBM traffic model, traffic mix of 50% VoIP and 50% video communications, and Num$_{STA}$ = 4. As the simulation result shows, the system performance increases when SNR values increase, so the maximum performance of 589 Mbps is achieved when SNR = 20 dB. This shows that the system performance increases when the channel quality improves with respect to the higher SNR values. This is due to the reduction of the frame error occurrence rate in the system. As the result shows, the proposed approach achieved the maximum performance in all channel conditions due to the employed dynamic adaptive aggregation selection scheme that maximizes the system throughput as compared to the All Agg FA (Baseline Approach). According to this experiment, Avg Num MPDUs FA contributed to the optimal aggregation strategy in SNR = 10 dB and 20 dB.

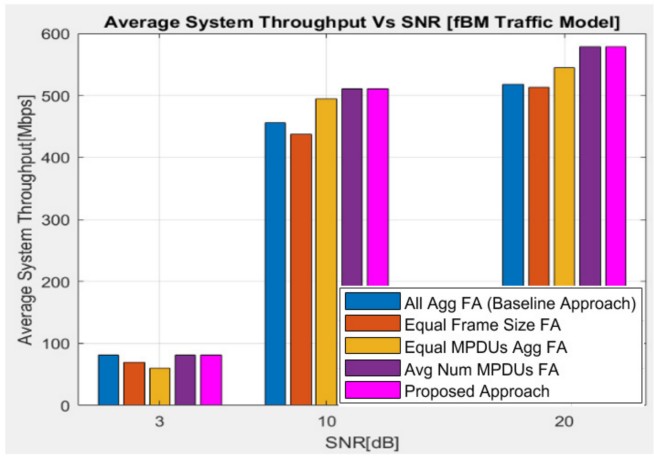

**Figure 4.** Average system throughput under varying SNR channel conditions.

We also perform a simulation experiment to examine the performance of system throughput under the effect of number of STAs, fBM traffic model, traffic mix of 50% VoIP and 50% video communications, and Num$_{STA}$ = 4. Figure 5 shows the performance of the proposed approach compared with different aggregation policies when the number of STAs changes from 2 to 4 and under SNR = 10 dB. According to the result, as the number of STAs increases, the system throughput performance increases. This is due to the fact that when the number of users in the system increases, the number of frames generated in the system increases. According to the result, Equal Num MPDUs FA contributed the maximum performance of 310 Mbps when Num$_{STA}$ = 2, and Ave Num MPDUs FA when Num$_{STA}$ = 3 and 4. This result shows that the aggregation policy used is affected by the number of users in the system. Due to the adaptive nature of the proposed algorithm, better performance is achieved in all conditions as compared to the All Agg FA (Baseline Approach). Then,

578 Mbps maximum performance is achieved when $Num_{STA} = 4$. A significant outcome of this experiment is that the number of users in a system could affect the determination of the optimal aggregation policy in the network. In this regard, the proposed approach achieved better performance by employing a dynamic adaptive aggregation selection scheme.

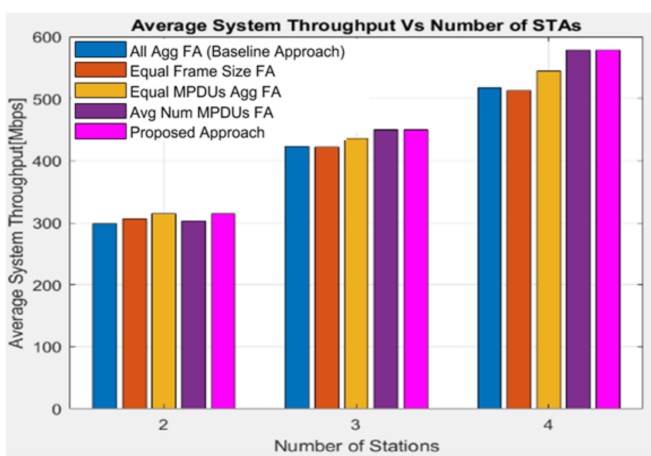

**Figure 5.** Performance of average system throughput as the number of STAs varies from 2 to 4.

### 5.3. Performance of System Throughput under Variable Traffic Data Types

The performance of system throughput under variable traffic data types considering different traffic mixes is examined in Figure 6. Since our analysis mainly focuses on aggregation, the traffic mix is performed in terms of frames (rather than the more common usage of bit rate). To achieve this, the traffic index type r is introduced to determine the traffic mix type in the simulation. Then, the traffic generator is invoked according to the index r. The two traffic types (VoIP and video) are generated independently as they have different packet rates and frame sizes. To differentiate between the different traffic mixes, we considered five values for r, where r = 1 is for 25% VoIP and 75% video frames; r = 2 is for 50% video, 25% VoIP frames; r = 3 is for 75% VoIP, and 25% video frames; r = 4 is only for VoIP frames; r = 5 is only for video. For example, if the simulation is set to r = 2, the system invokes both the VoIP and video traffic generators sequentially. Then, to maintain a 50%:50% traffic ratio of r = 2, the ratio analyzer computes the total number of traffic streams generated for a user, i.e., T= $\sum$ (VoIP + video). The packet rate for VoIP and video is then adjusted according to the ratio specified for r = 2, i.e., T × 50% for VoIP and the test T × 50% for video.

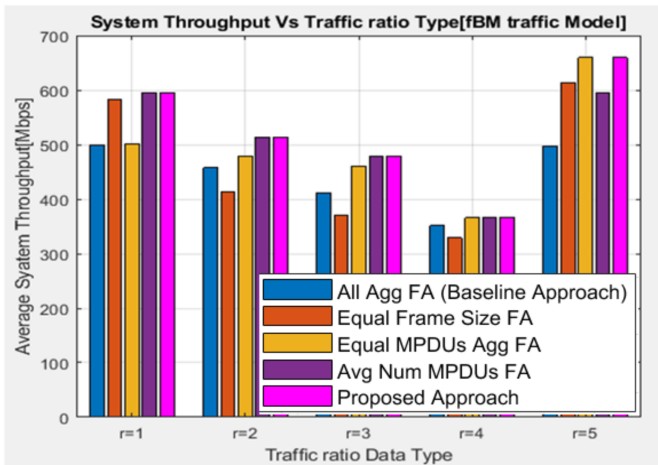

**Figure 6.** Performance of average system throughput under varying mixes of VoIP and video traffic.

The experiment in Figure 6 demonstrates the performance of the proposed approach under the effects of variable traffic types. SNR = 10 dB and $\text{Num}_{\text{STA}}$ = 4 are considered in this experiment. The result from this experiment indicates that since the actual network traffic varies considerably with different traffic usage patterns, one type of aggregation strategy may not always achieve the maximum system throughput performance. For instance, when r = 5, Equal Num MPDUs contribute to the maximum performance, whereas All Agg FA (Baseline Approach) is the poorest. According to this result, the proposed adaptive approach offers a better performance in selecting the optimal aggregation strategy that maximizes the system throughput in all traffic scenarios. Under this channel condition, the maximum performance of 650 Mbps is achieved when the traffic type is all video (r = 5). This is because the video traffic data accommodate more frames with a size of 1000 Mbytes, compared to 100 Kbytes for VoIP traffic which is the worst performance. The significant outcome of this experiment depicted that the variable traffic types in a network could affect the determination of the optimal aggregation policy that would maximize the system throughput performance. In this regard, the proposed adaptive aggregation selection scheme achieved a significant performance in all traffic ratio data type conditions.

### 5.4. Performance of System Throughput Performance with Increasing Traffic Load

In Figures 7–9, the system throughput performance with increasing system traffic load, SNR = 10 dB, and $\text{Num}_{\text{STA}}$ = 4 for 50% video traffic and 50%VoIP traffic types is evaluated. Different traffic models such as Pareto, Weibull, and fBM are considered in this experiment. Average offered traffic [Mbps] is the average data frame size generated in the system per unit of time. Average system throughput [Mbps] is defined as the average data rate at which the AP can successfully transmit to all STAs.

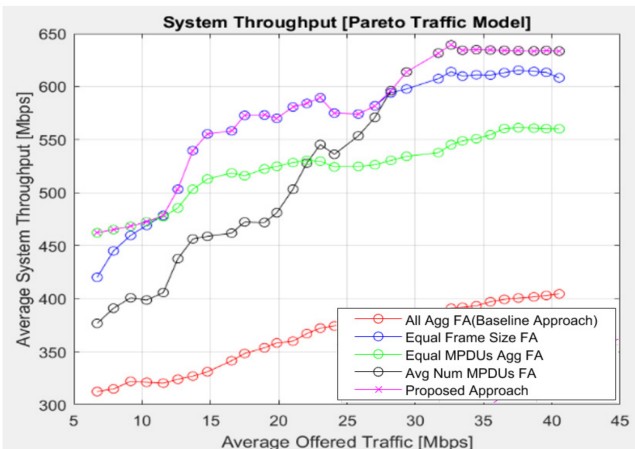

**Figure 7.** Performance of average system throughput with increasing offered traffic load using Pareto traffic model.

Figure 7 shows the performance using the Pareto traffic model. As the results show, a particular aggregation strategy cannot achieve maximum performance when the traffic load increases. This result shows that different aggregation policies perform differently due to the effects of the traffic pattern in this channel condition. As the result shows, the combination of Equal Frame Size FA, Avg Num MPDUs FA, and Equal Num MPDUs FA aggregation policies contributed to the maximum performance in this traffic model. However, due to the employed adaptive aggregation selection procedure in the proposed approach, the maximum system performance is achieved throughout the simulation time. A maximum of 647 Mbps is achieved, compared to All Agg FA (Baseline Approach) which had the lowest, at 410 Mbps.

The simulation result in Figure 8 shows the performance of the system throughput using Weibull traffic load. In this traffic model, Equal Num MPDUs FA contributed to the optimal aggregation policy when the traffic load increased, particularly after 5 Mbps.

However, the proposed approach achieved the maximum performance of 755 Mbps, compared to all approaches throughout the stimulation time employing the dynamic adaptive aggregation selection scheme. All Agg FA (Baseline Approach) achieved the poorest performance.

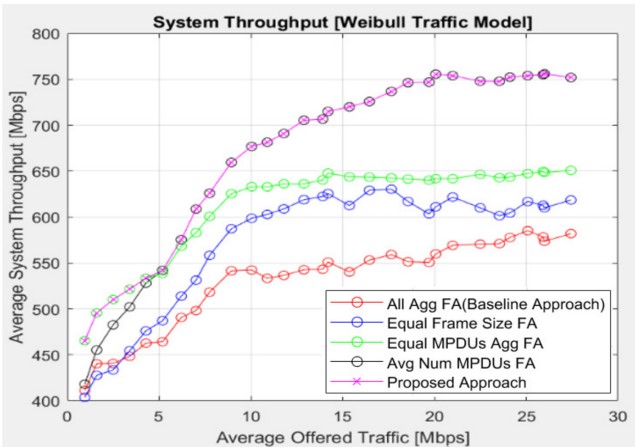

**Figure 8.** Performance of average system throughput with increasing offered traffic load using the Weibull traffic model.

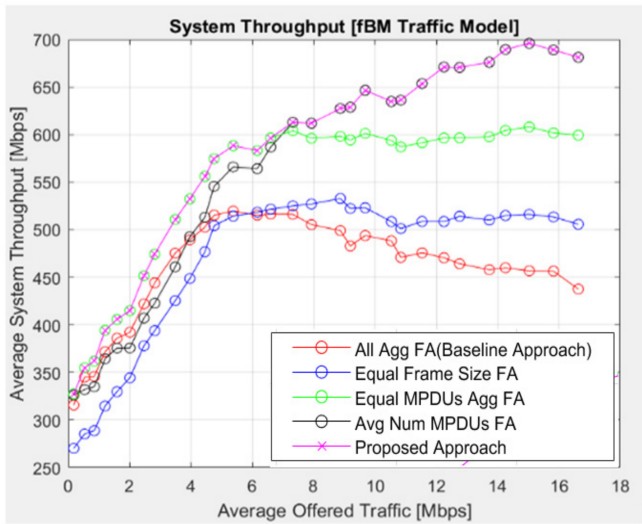

**Figure 9.** Performance of average system throughput with increasing offered traffic load using fBM traffic model.

Similarly, the simulation result in Figure 9 demonstrates the performance of system throughput using fBM traffic model. As the results show, Equal MPDUs Agg FA contributed the maximum performance of up to 6 Mbps Average traffic load, and Avg Num MPDUs FA contributed beyond 6 Mbps. However, the proposed algorithm achieved the maximum performance of 700 Mbps by adaptively selecting the optimal aggregation policy throughout the simulation time, thus achieving better performance compared to the baseline All Agg FA approach.

In general, as the results demonstrated in Figures 7–9, the performance of system throughput is affected due to heterogeneous traffic models. Different frame aggregation policies perform differently in different traffic models. In this regard, as the results show in all figures, such results are mainly used to illustrate how the performance of the proposed approach increases with the dynamic effects of different aggregation approaches proposed in the system. According to the results, the proposed adaptive aggregation approach

is significant to realize the maximum system throughput performance with respect to different traffic models in the downlink MU-MIMO channel.

### 5.5. Performance of Space Channel Time with Increasing Traffic Load

Figures 10–12 show the performance of the proposed approach evaluated with respect to space channel time, with increasing system traffic load, under the effect of different traffic models such as Pareto, Weibull, and fBM, with SNR = 10 dB, $Num_{STA}$ = 4, and 50% video and 50% VoIP traffic data. The system-wasted space channel time (SP) is the ratio of the sum of wasted space channel time to the sum of all channel time (i.e., data transmission time and space channel time). It is evaluated according to Equation (5).

$$Space_i = Max(TXData) - TXData_i$$

$$SP(\%) = \frac{\sum_{i=1}^{4} Space_i}{\sum_{i=1}^{4} Space_i + \sum_{i=1}^{4} TXData_i} \times 100 \tag{5}$$

where:

- $Max(TXData)$ is the maximum transmission duration among the parallel streams;
- $Space_i$ is wasted space channel time of $STA_i$;
- $TXData_i$ is the data transmission time of $STA_i$;

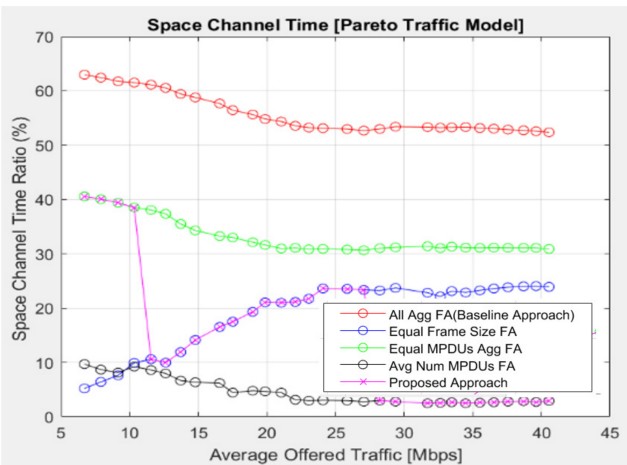

**Figure 10.** Performance of space channel time ratio with increasing offered traffic load using the Pareto traffic model.

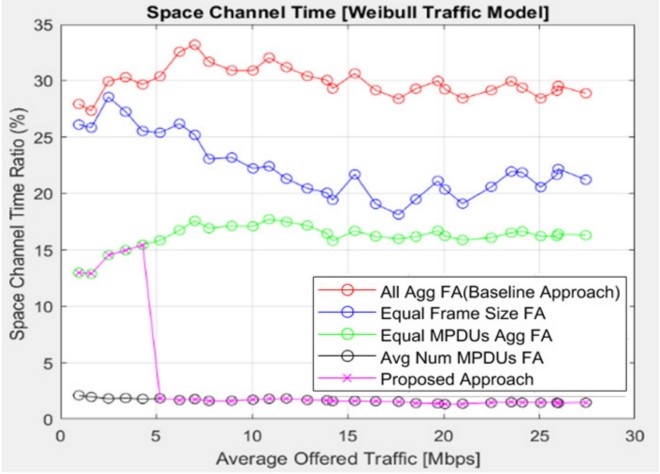

**Figure 11.** Performance of space channel time ratio with increasing offered traffic load using Weibull traffic model.

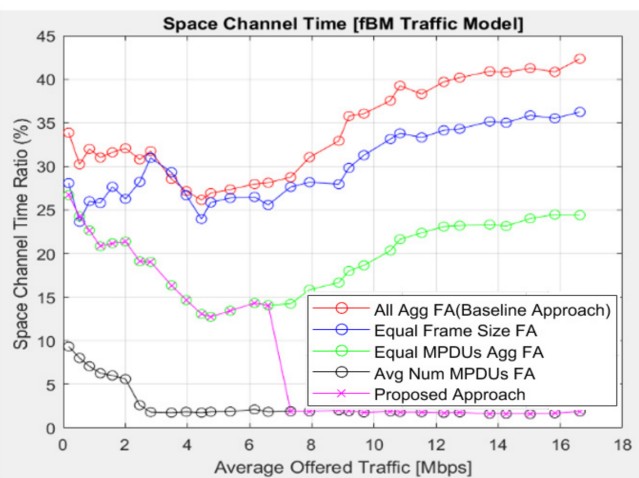

**Figure 12.** Performance of space channel time ratio with increasing offered traffic load using fBM traffic model.

The result in Figure 10 shows the performance of wasted space channel time using the Pareto traffic model. According to the result, the proposed approach significantly decreases below 5% as the traffic load increases when compared with the baseline All Agg FA algorithm, while it slightly increases at the beginning and middle of the simulation. This is because of the employed dynamic adaptive aggregation selection scheme employed by the proposed approach. This result indicates that the proposed approach is significant to reduce wasted space channel time by minimizing the occurrence of the high variability of data transmission duration among spatial streams.

The result in Figure 11 also shows the performance of space channel time using the Weibull traffic model. The smaller space channel time wastage of less than 2% is achieved by the proposed approach due to the adaptive aggregation strategy it has employed. Space channel time wastage increases when the traffic variation among spatial streams increases in the downlink MU-MIMO channel. This leads the performance of All Agg FA (Baseline Approach) archives to the worst 33% wastage of space channel time, since it promotes a longer aggregated frame size strategy [11]. As the result shows, the performance of space channel time in the Weibull traffic model is better than Pareto due to the highest bursty traffic pattern in Pareto compared to Weibull. Therefore, this result indicates that the proposed approach performs better in the Weibull traffic model.

Figure 12 shows the performance of wasted space channel time using the fBM traffic model. As the result shows, different aggregation policies achieve different space channel time performance. However, the proposed approach achieved the minimum performance particularly after the offered traffic load was beyond 7 Mbps. Therefore, better performance is achieved compared with the baseline All Agg FA approach. According to the results, the proposed approach achieved better performance in the fBM and Weibull traffic models than in Pareto.

In general, the simulation results in Figures 10–12 demonstrated that the system space channel time is affected by the traffic pattern in the network. As the results show, they mainly illustrate how the performance of the proposed approach adaptively reduces space channel time in the system with the dynamic aggregation schemes employed in the system. As the results show, better performance is achieved in the Weibull traffic model and fBM than the Pareto one since Pareto is highly bursty. However, the proposed approach achieved better performance in minimizing the space channel time wastage in all scenarios. According to the results shown in Figures 7–12, the proposed algorithm has the potential to achieve joint optimization capabilities by maximizing system throughput in minimizing space channel time.

## 6. Conclusions and Future Works

In this paper, we proposed a dynamic adaptive frame aggregation selection algorithm in the WLAN downlink MU-MIMO channel. The main contribution of this paper is to achieve an efficient adaptive frame aggregation algorithm for downlink MU-MIMO transmission in terms of system throughput performance and space channel time by extending our previous work. The proposed scheme examines the challenges of heterogeneous traffic demand among streams in the downlink MU-MIMO channel under the condition of channel error. A novel criterion was proposed to select the optimal frame aggregation policy, which allowed the optimal wireless frame setting to be determined. The experiment was conducted in two steps. Step 1 performed the dynamic optimal aggregation policy selection strategy as per the channel condition, traffic pattern, and number of stations in the network. Step 2 then performed the optimal wireless frame construction that would be transmitted in the wireless channel in adopting the optimal aggregation policy obtained from Step 1. The performance of our algorithm was evaluated through system-level simulation using MATLAB programming. Furthermore, the proposed adaptive algorithm not only achieved the optimal system throughput in minimizing wasted space channel time but also achieved a good performance under the effects of different channel conditions, different traffic models such as Pareto, Weibull, and fBM, and number of users. A traffic mix of VoIP and video data were used in the experiment. From the results, it can be seen that the proposed algorithm significantly improves system performance compared to that of the baseline FIFO algorithm. Under further work, we plan to investigate the effects of delay on the performance of our algorithm in both streams of the uplink and downlink MU-MIMO channel of IEEE802.11 ax networks considering real traffic data and multiple user scenarios. In addition, we will extend our approach by employing an adaptive machine learning approach.

**Author Contributions:** We would like to thank all authors for their contributions and the successes of this manuscript. Conceptualization, L.K., M.D., and J.D.; Data curation, L.K.; Investigation, L.K.; Methodology, L.K.; Software, L.K.; Supervision, M.D., J.D., and J.C.; Validation, M.D.; Writing—original draft, L.K.; Writing—review and editing, M.D. Moreover, we would like to thank the editors and anonymous reviewers of this manuscript. All authors have read and agreed to the published version of the manuscript.

**Funding:** This research received no external funding.

**Data Availability Statement:** All data are available to any researcher upon request.

**Conflicts of Interest:** The authors declared no potential conflicts of interest concerning the research, authorship, and/or publication of this article.

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
