# Peer review of "Performance of an Adaptive Aggregation Mechanism in a Noisy WLAN Downlink MU-MIMO Channel"

_electronics, doi:10.3390/electronics11050754_

Round 1
Reviewer 1 Report
The idea used in research is clearly specified. However, it needs improvement in English to make it more presentable. Further, the techniques applied needs to be more precisely expressed as it is difficult how step1 and step 2 are used together. Channel error is assumed in terms of SNR, but in wireless, there may be other reason for channel error such as fading and shadowing, etc. How is this condition of channel error is addressed?
The following suggestions are recommended to improve the quality of the paper:
- English of abstract should be improved
- Aggregation process should be more precisely explained.
- Figure 2 should be explained. space 1; space 2; space 3 as shown in figure is clear.
- There are too many graphs/figures which can be reduced, if possible.
- Conclusion must include findings of the research.
- Other conditions of channel error should either be considered or justification for not considering it should be given.
Author Response
Dear Reviewer,
Thanks so much for all the fruitful comments and suggestions you gave us for a better-presented manuscript. Here I sent you the responses according to your comments and moreover, the corrected manuscript is also attached with this email, please find it.
1. English of abstract should be improved
Response: revised and can be seen from Line 10-28
2. Aggregation process should be more precisely explained.
Response: Revised and can be seen from Line 280- 290
3. Figure 2 should be explained. space 1; space 2; space 3 as shown in figure is clear.
Response: Revised and can be seen from Line 213-217
4. There are too many graphs/figures which can be reduced, if possible.
Response: Figure 3 and Figure are merged in one to reduce number of figures. A new Figure 3 illustrated and can be seen from Line 345-346
5. Conclusion must include findings of the research.
Response: Revised and can be seen from Line 574-597
6. Other conditions of channel error should either be considered or justification for not considering it should be given.
Response: Revised. Can be seen from Line 239-243 and 375 -391
With Kindest Regards;
Lemlem
Reviewer 2 Report
Dear Editors, Well written paper but can be improved incorporating the following comments: 1. In Abstract, referring to earlier work (line 14-16), I am not which earlier work. It is confusing. 2. Line 66-67, define the terminology when using first time- example is SNR but applied all terminology in the paper. 3. I think it is simulation setup not the experimental - Figure 3 and Figure 4. Need to update in the whole paper. 4. Table 1: # of Stations are 2-4, which is not realistic scenarios. You need to do at least 10-15 to simulate the realistic situation. 5. Figure 8-13, you need to include detailed analysis why the proposed approach does have own results but following other approach at certain points. This analysis will provide more insight on the results. Compare Figure 8 and 11, why transition are not happening at same points? Same applies to other figures. Good luck.Author Response
Dear Reviewer,
Thanks so much for all the fruitful comments and suggestions you gave us for a better-presented manuscript. Here I sent you the responses according to your comments and the corrected manuscript is also attached with this email, please find it.
- In Abstract, referring to earlier work (line 14-16), I am not which earlier work. It is confusing.
Response: As we mentioned in many places in the manuscript, the main contribution of this manuscript is to extend our earlier work which is specified on the reference number [11].
- Line 66-67, define the terminology when using first time- example is SNR but applied all terminology in the paper.
Response: Revised and can be seen from Line 240
- I think it is simulation setup not the experimental - Figure 3 and Figure 4. Need to update in the whole paper.
Response: Revised. Line 346. And it is also updated in the whole manuscript.
- Table 1: # of Stations are 2-4, which is not realistic scenarios. You need to do at least 10-15 to simulate the realistic situation.
Response: Revised and can be seen from Line 187
- Figure 8-13, you need to include detailed analysis why the proposed approach does have own results but following other approach at certain points. This analysis will provide more insight on the results. Compare Figure 8 and 11, why transition are not happening at same points? Same applies to other figures.
Response: Revised and can be seen from Line 510-514, Line 564-567. Basically, the results are mainly to illustrate how the proposed approach adaptively increases with the change of different aggregation approaches.
Kindest Regards,
Lemlem